# Methylated Reprimo Cell-Free DNA as a Non-Invasive Biomarker for Gastric Cancer

**DOI:** 10.3390/ijms26073333

**Published:** 2025-04-03

**Authors:** María José Maturana, Oslando Padilla, Pablo M. Santoro, Maria Alejandra Alarcón, Wilda Olivares, Alejandro Blanco, Ricardo Armisen, Marcelo Garrido, Edmundo Aravena, Carlos Barrientos, Alfonso Calvo-Belmar, Alejandro H. Corvalán

**Affiliations:** 1Department of Hematology and Oncology, Faculty of Medicine, Pontificia Universidad Católica de Chile, Portugal 61, Santiago 8330023, Chile; mariaj.maturana@gmail.com (M.J.M.); p.santoro.v@gmail.com (P.M.S.); titalarcon@gmail.com (M.A.A.); wdolivar@uc.cl (W.O.); drmgarrido@gmail.com (M.G.); 2School of Public Health, Faculty of Medicine, Pontificia Universidad Católica de Chile, Santiago 8330023, Chile; opadilla@uc.cl; 3Centro de Genética y Genómica, Instituto de Ciencias e Innovación en Medicina (ICIM), Facultad de Medicina Clínica Alemana Universidad del Desarrollo, Santiago 7550000, Chile; ablanco@udd.cl (A.B.); rarmisen@udd.cl (R.A.); 4Instituto Chileno Japones de Enfermedades Digestivas, Hospital Clinico San Borja Arriaran, Servicio Salud Metropolitano Central, Santiago, Chile and Fundación Arturo López Pérez, Santiago 8360160, Chile; edmundoaravena@yahoo.de (E.A.); barrientosc@falp.org (C.B.); 5Hospital Dr. Sotero del Rio, Servicio Salud Metropolitano Sur-Oriente, Santiago 8207257, Chile; acalvobelmar@yahoo.es

**Keywords:** gastric cancer, biomarkers, liquid biopsy, non-invasive diagnosis, cancer screening, cancer prevention, methylated *RPRM*, cfDNA

## Abstract

Restrictions resulting from the COVID-19 pandemic abruptly reversed the slow decline of the diagnosis and mortality rates of gastric cancer (GC). This scenario highlights the importance of developing cost-effective methods for mass screening and evaluation of treatment response. In this study, we evaluated a non-invasive method based on the circulating methylated cell-free DNA (cfDNA) of Reprimo (*RPRM*), a tumor suppressor gene associated with the development of GC. Methylated *RPRM* cfDNA was analyzed in three de-identified cohorts: Cohort 1 comprised 81 participants with GC and 137 healthy donors (HDs); Cohort 2 comprised 27 participants with GC undergoing gastrectomy and/or chemotherapy analyzed at the beginning and after three months of treatment; and Cohort 3 comprised 1105 population-based participants in a secondary prevention program who underwent esophagogastroduodenal (EGD) endoscopy. This cohort includes 180 normal participants, 845 participants with premalignant conditions (692 with chronic atrophic gastritis [AG] and 153 with gastric intestinal metaplasia/low-grade dysplasia [GIM/LGD]), 21 with high-grade dysplasia/early GC [HGD/eGC], and 59 with advanced GC [aGC]). A nested case-control substudy was performed using a combination of methylated *RPRM* cfDNA and pepsinogens (PG)-I/II ratio. The dense CpG island of the promoter region of the *RPRM* gene was bisulfite sequenced and analyzed to develop a fluorescence-based real-time PCR assay (MethyLight). This assay allows the determination of the absolute number of copies of methylated *RPRM* cfDNA. A targeted sequence of PCR amplicon products confirmed the gastric origin of the plasma-isolated samples. In Cohort 1, the mean value of GCs (32,240.00 copies/mL) was higher than that of the HD controls (139.00 copies/mL) (*p* < 0.0001). After dividing this cohort into training–validation subcohorts, we identified an area under the curve of 0.764 (95% confidence interval (CI) = 0.683–0.845) in the training group. This resulted in a cut-off value of 87.37 copies/mL (sensitivity 70.0% and specificity 80.2%). The validation subcohort predicted a sensitivity of 66.67% and a specificity of 83.33%. In Cohort 2 (monitoring treatment response), *RPRM* levels significantly decreased in responders (*p* = 0.0042) compared to non-responders. In Cohort 3 (population-based participants), 18.9% %, 24.1%, 30.7%, 47.0%, and 71.2% of normal, AG, GIM/LGD, HGD/eGC, and aGC participants tested positive for methylated *RPRM* cfDNA, respectively. Overall sensitivity and specificity in distinguishing normal/premalignant conditions vs. GC were 65.0% (95% CI 53.52% to 75.33%) and 75.9% (95% CI 73.16% to 78.49%), respectively, with an accuracy of 75.11% (95% CI 72.45% to 77.64%). Logistic regression analyses revealed an OR of 1.85 (95% CI 1.11–3.07, *p* = 0.02) and an odds ratio (OR) of 3.9 (95% CI 1.53–9.93, *p* = 0.004) for the risk of developing GIM/LGD and HGD/eGC, respectively. The combined methylated *RPRM* cfDNA and PG-I/II ratio reached a sensitivity of 78.9% (95% CI 54.43% to 93.95%) and specificity of 63.04% (95% CI 52.34% to 72.88%) for detecting HGD/eGC vs. three to six age- and sex-matched participants with premalignant conditions. Our results demonstrate that methylated *RPRM* cfDNA should be considered a direct biomarker for the non-invasive detection of GC and a predictive biomarker for treatment response.

## 1. Introduction

Global diagnosis and mortality rates of gastric cancer (GC) 1.1 million and 770,000 per year, respectively—were slowly declining prior to the COVID-19 pandemic [1]. Restrictions resulting from this health crisis abruptly reversed this decline, causing patients to present with more severe stages at diagnosis and worse survival outcomes [2,3]. This new scenario not only models the planning capacity of the health system but highlights the importance of developing screening programs for the early detection of GC [4].

Although both mass and opportunistic screening have reduced GC mortality, their implementation cost represents a major challenge for public health systems [5]. In this scenario, non-invasive approaches represent a cost-effective alternative and may be applied to evaluate treatment response [6].

A growing body of evidence has shown that the measurement of circulating cell-free DNA (cfDNA) may be a novel approach for non-invasive detection of a variety of tumor types [7,8]. One such approach is methylated cfDNA, an epigenetic alteration that inactivates tumor suppressor genes at the promoter region [9]. Methylated cfDNA biomarkers, such as the glutathione S-transferase gene (GSTP1) for prostatic cancer [10] and Sept9 for colorectal carcinoma [11], are well-established examples of this approach [12]. Several biomarkers have been identified in GC [13], but they have yet to be tested simultaneously in screening and treatment response. To address this gap, we evaluated one of the most promising candidates, the methylated cfDNA of Reprimo (*RPRM*) [14].

*RPRM* is a tumor suppressor gene involved in the regulation of the cell cycle [15] as well as apoptosis [16]. We have found that the loss of *RPRM* expression is associated with DNA hypermethylation of its promoter region [17]. It has been described, both by us and by other researchers, in tissue samples from a variety of tumors (for a review, see [18]). The DNA hypermethylation of *RPRM* has exclusively been described in precancerous tissues in the stomach [19].

Our results support the idea that the methylated cfDNA of *RPRM* may be a valuable biomarker for the non-invasive detection of GC as well as treatment response prediction.

## 2. Results

### 2.1. RPRM Promoter Methylation in Gastric Cancer Tissues

We analyzed the dense CpG island of the promoter region of the *RPRM* gene (National Center for Biotechnology Information [NCBI] NM_019845/Gene ID: 56475), located from nucleotides −207 to +309 relative to the transcription start site (TSS) [17]. This region was bisulfite sequenced from peripheral blood mononuclear cells (PBMC) from healthy donors (HDs) (*n* = 2) and tissues from chronic atrophic gastritis (AG) (*n* = 3) and GC (*n* = 6). Low methylation levels (< 10%) were found in the HDs, while the AG cases demonstrated intermediate levels of hypermethylation (50.4%), and the GC tissues showed the highest levels (65.5%) of CpG island methylation (Figure 1). Based on these findings, we converted our previously reported MSP [14] into a fluorescence-based real-time PCR assay (MethyLight) which allows for the calculation of the absolute number of copies of the methylated DNA after sodium bisulfite conversion [20].

### 2.2. RPRM Promoter Methylation in Paired Tissue and Plasma Gastric Cancer Samples

To validate the capacity of this novel assay to detect the presence of *RPRM* promoter DNA methylation as a surrogate cfDNA biomarker for GC, 20 paired tumor and plasma tissue samples were analyzed using MethyLight. This analysis resulted in 19/20 (95%) positive cases. A targeted sequence of the PCR amplicon products was performed on three paired samples to confirm the gastric origin of the plasma-isolated samples. As shown in Figure 2, two identical clones were identified in one of these samples. In the other cases, four and five identical clones were identified, respectively (Appendix A).

### 2.3. Characterization of Methylated RPRM Cell-Free DNA as a Biomarker for Non-Invasive Detection of Gastric Cancer (Cohort 1)

The gastric origin of methylated *RPRM* cfDNA suggests that *RPRM* promoter methylation could be used as a biomarker for non-invasive assessment of GC as well as monitoring treatment response. To evaluate the first of these potentialities, methylated *RPRM* cfDNA was prospectively collected and assessed in 81 GC participants from the blood plasma samples obtained prior to esophagogastroduodenoscopy (EGD) and from 137 HD controls. Although the age of the GC group was older than that of the control group (*p* < 0.0001), the proportion of male to female participants was similar (ratio 1.9:1 vs. 2:1, respectively). The ratio of intestinal- to diffuse-type GC was 2.5:1, and the ratio of early to advanced stages was 1:4.3. Due to the age discrepancy between the cancer cases and the HD controls (*p* < 0.0001), we performed a pairwise comparison by quantile regression models. This analysis indicated that the values of methylated *RPRM* cfDNA do not increase according to age. These findings suggest that age differences did not affect our results (Appendix A). A total of 73% (59/81) of the GC participants and 31% (42/137) of the HD controls were positive for methylated *RPRM* cfDNA. As shown in Figure 3A, the mean value of the GC cases (32,240.00 copies/mL, SD = 157,700.00 copies/mL) was higher than that of the HD controls (139.00 copies/mL, SD = 657.00 copies/mL). These differences were statistically significant (*p* < 0.0001; Mann–Whitney test). To identify the best sensitivity and specificity of the methylated *RPRM* cfDNA for non-invasive assessment, patients and controls were randomly divided through a training–validation approach and analyzed in a blind fashion. As shown in Figure 3B, the ROC curve of the training group (60 GC participants and 101 HD controls) demonstrated an AUC of 0.764 (95% confidence interval (CI) = 0.683–0.845) to determine the optimal cut-off value of 87.37 copies/mL (sensitivity 70.0%, and specificity 80.2%). The prediction accuracy was assessed in the validation group (21 GC and 36 HDs), which showed a sensitivity of 66.67% and a specificity of 83.33% (Appendix A).

### 2.4. Characterization of Methylated RPRM Cell-Free DNA as a Biomarker for Monitoring Gastric Cancer Treatment Response (Cohort 2)

To evaluate the role of methylated *RPRM* cfDNA in monitoring GC treatment response, *RPRM* levels were measured in plasma samples from 27 patients undergoing total gastrectomy, palliative chemotherapy, or perioperative chemotherapy (Appendix A—Cohort 2). Measurements were performed at the beginning of treatment (T0) and again 3 months after (T1). The Response Evaluation Criteria in Solid Tumours (RECIST) score score was used for individual evaluation of the treatment response. The analysis of methylated *RPRM* cfDNA levels between T0 and T1 revealed a significant decrease in the responders (*p* = 0.0042; Wilcoxon paired signed rank test) (Figure 4). Among non-responders who showed stable or progressive disease, no significant differences were observed (*p* = 0.54; Wilcoxon matched pairs signed rank test).

### 2.5. Evaluation of Methylated RPRM Cell-Free DNA Across the Premalignant Conditions and Gastric Cancer (Cohort 3)

Having shown that methylated *RPRM* cfDNA can be a biomarker for non-invasive assessment and monitoring treatment response, we evaluated its capacity to distinguish premalignant conditions of the stomach. To accomplish this, 1105 consecutively referred symptomatic patients were accrued for EGD procedures and analyzed in a blind fashion. MethyLight detected methylated *RPRM* in cfDNA extracted from the blood plasma sample prior to EGD. The distribution of the population study is shown in Appendix A—Cohort 3. Participants with normal EGD were considered the control group. This group represents 16.3% (*n* = 180) of the sample, with a mean age of 57 years old (range 38–85). The predominant group (62.6%, *n* = 692 of the sample) was comprised by AG cases with an average age of 58 years old (range 36–88). The cases diagnosed with gastric intestinal metaplasia (GIM) (*n* = 136) and low-grade dysplasia (LGD were grouped together due to the small number in the latter group (LGD = 13), accounting for 13.8% of the series. The mean age of this group was 63 years old (range 38–92). The early GC (eGC) and high-grade dysplasia (HGD) cases represent 1.9% of the sample and were also grouped together due to the small number of patients in both groups (eGC = 18 and HGD = 3). The mean age of this group was 67 years old (range 47–95) (*p* < 0.0001). The advanced GC (aGC) cases accounted for 5.3% of the series, with an average age of 65 years old (range 17–85). Women were more prevalent in the control and AG groups (81.7% and 69.1%) (*p* < 0.0001) and conversely, men were prevalent in the aGC group (64.4%) (*p* < 0.0001) (Appendix A—Cohort 3). *H. pylori* was evaluated in a subset of premalignant and tumor lesions (*n* = 172, 15.56%), with positive results in 69.7% (62/89), 60.9% (42/69), and 28.6% (4/14) of AG, GIM/LGD, and eGC/HGD, respectively (*p* = 0.012 Pearson’s) (Appendix A—Cohort 3).

A total of 18.9% of the normal patients tested positive for methylated *RPRM* cfDNA, compared to 24.1% of the AG patients and 30.7% of the GIM/LGD patients. A total of 47.0% (*n* = 10) of the HGD/eGC cases and 71.2% (*n* = 42) of the aGC cases, respectively, tested positive for methylated *RPRM* cfDNA. As shown in Table 1, the overall sensitivity (true positive rate) was 65.0% (95% CI 53.52% to 75.33%), and the specificity (or false positive rate) was 75.9% (95% CI 73.16% to 78.49%). Moreover, the positive predictive value was estimated at 17.39% (95% CI 14.78–20.36%), while the negative predictive value was 96.53% (95% CI 95.36–97.40%).

Considering these parameters, the overall accuracy of methylated *RPRM* cfDNA to distinguish GC from normal subjects and premalignant patients was 75.11% (95% CI 72.45% to 77.64%). These values were more accurate for aGC than HGD/eGC.

Logistic regression analyses revealed that methylated *RPRM* cfDNA was not capable of distinguishing the normal from the AG subjects (Table 2). However, it was able to identify the risk of developing GIM/LGD (odds ratio (OR) 1.85 (95% CI 1.11–3.07), *p* = 0.02), and it was even more accurate at identifying HGD/eGC (OR 3.9 (95% CI 1.53–9.93), *p* = 0.004) and aGC (OR 10.61 (95% CI 5.409–20.85), *p* < 0.0001) (Table 3). After adjusting for sex and age, only HGD/eGC (OR 3.23 (95% CI 1.15–9.11) *p* < 0.026) and aGC (OR 9.79 (95% CI 4.43–21.61) *p* < 0.0001) were accurately identified by methylated *RPRM* cfDNA (Table 2).

A stratified analysis by lesion grade (adjusted by age and sex reveals that these variables do not influence the accuracy of methylated *RPRM* cfDNA (Appendix A—Cohort 3). The rate of detection of *H. pylori* among methylated *RPRM* cfDNA positive cases was significatively lower among the HGD/eGC cases (1/7, 14.3%) than that of the AG cases (14/17, 82.4%) and the GIM/LGD cases (15/19; 78.9%) (*p* = 0.004 Fisher’s exact test). Taken together, the detection of methylated *RPRM* cfDNA in a blood sample highly correlates with the presence of HGL/eGC and aGC.

To improve our results for the non-invasive diagnosis of the early stages of GC, we combined methylated *RPRM* cfDNA with a PG-I/II ratio. Twenty-one HGD/eGC cases were matched with three to six age and sex controls from each lesion of the premalignant conditions (AG and GIM/LGD). As shown in Table 3, the combined assay reached a sensitivity of 78.9% (95% CI 54.43% to 93.95%) and specificity of 63.04% (95% CI 52.34% to 72.88%) with a positive and negative predictive value of 30.61% (95% CI 23.65% to 38.59%) and 93.55% (95% CI 85.69% to 97.23%), respectively. In this series we found 34 (69.4%) false positives and 4 (6.8%) false negatives.

Considering these results, the overall accuracy of the combined use of the methylated *RPRM* cfDNA and the PG-I/II ratio to distinguish HGD/eGC from precancerous lesions was 65.77% (95% CI 56.16% to 74.51%).

## 3. Discussion

The slow decline in the global diagnosis and mortality rates of GC abruptly reversed in 2020 due to the COVID-19 pandemic [21]. To address this dire situation, biomarkers for the non-invasive diagnosis of tumors, which must be included in health and/or metastases [9], has emerged as a novel biomarker system planning capacity [22,23]. Circulating cell-free DNA (cfDNA), derived from primary tumors for the non-invasive detection of cancer, is now one of the most extensively studied areas in translational research [7,8,24,25]. A consolidation of this expanding field is the LiqBioer, a manually curated database of cancer biomarkers in body fluids [26].

In this study, we developed a novel fluorescence-based real-time PCR assay based on the methylation of the promoter region of *RPRM*, a well-established tumor suppressor gene and recognized biomarker for GC [14,17,27,28,29]. Our assay can calculate the absolute number of bisulfite-converted DNA molecules using a hybridization probe and a standard curve. The gastric origin of this biomarker in plasma was confirmed by the targeted sequence of PCR amplicon products. This finding aligns with large-scale genomic profiling of the cfDNA that reproduces the genomic landscape of driver mutations found in primary tumor tissues [30]. The assessment of methylated *RPRM* cfDNA as a non-invasive approach revealed that, despite the age discrepancy between cases and controls, our predicted cut-off reaches sensitivities and specificities replicated in the validation cohort. Age discrepancies are frequently observed in screening studies and might be a limitation of a cancer screening program [31]. In treatment response, methylated *RPRM* cfDNA revealed a significant decrease among responders, but not in the group of non-responders. In particular, the observation that methylated *RPRM* cfDNA levels decreased dramatically after three months of treatment also provides new opportunities to detect residual disease in the GC space [32]. Our results of similar clones in tumor and plasma aligned with the high correlation of tumor mutational burden found by Maron et al. [33] in paired tissue-cfDNA from patients with MSI-high tumors. An ongoing prospective observational study focusing on this issue (PLAGAST, NCT-02674373) could provide more insight into this area.

We evaluated our assay on 1105 consecutively symptomatic patients representing all steps of the premalignant conditions of the GC cascade. This analysis shows that *RPRM* methylated cfDNA correlates with the presence of GC with an overall accuracy of 75.11% (95% CI 72.45% to 77.64%). As expected, these values were more accurate for aGC than eGC/HGD. In addition, logistic regression analyses revealed that methylated *RPRM* cfDNA could progressively identify advanced lesions of the cascade or GC at early or advanced stages. This finding was even more accurate when the analysis was adjusted by sex or age. In cases of eGC, the combined analysis of *RPRM* methylated cfDNA and the PG-I/II ratio yielded sensitivities and specificities higher than those achieved by each one alone. The combined result warrants further exploratory research. Of note, the prevalence of infection decreased across the premalignant conditions.

Pepsinogens (PGI < 70 or PG-I/II ratio < 3) are the longest standing candidates for the non-invasive detection of GC and its precursor lesions. Multiple studies have shown a moderate sensitivity and specificity for these biomarkers (for a review see [34]). Combining PGs with *H. pylori* detection, known as the ABC method [35], can improve their performance, particularly in patients in which *H. pylori* has been eradicated [36]. Trefoil factor 3 (TFF-3) belongs to a secretory peptide family that maintains gastric mucosal integrity. TFF3 has reached 80.9% sensitivity and 81.0% specificity as a non-invasive biomarker for GC [37]. However, serum levels of TFF3 did not respond after treatment, meaning they might not be useful for monitoring response. Carcinoembryonic antigen (CEA), carbohydrate antigen 19-9 (CA19-9), and carbohydrate antigen 72-4 (CA72-4) have low sensitivity and specificity [38]. A single study shows that the combined use of these biomarkers reaches 67% and 89% sensitivity and specificity, respectively [39]. Recently, So et al. [40] evaluated a serum-based biomarker signature of 12 microRNAs. In a large prospective cohort (*n* = 4566 participants) this panel distinguished the GCs from the HDs with sensitivity of 87% and specificity of 68.4%. These results are promising; however, RNAs are prone to degrade faster than DNA-based assays and levels can vary rapidly depending on gastric condition (fasting or not fasting) [41]. Both conditions are important considerations for the implementation of a clinical test.

A limitation of this study in Cohort 1 was the age differences between the GC and control groups (*p* < 0.0001). A pairwise comparison by quantile regression models showed differences according to age did not affect our results. In Cohort 2, the observed decrease of methylated *RPRM* cfDNA after treatment may be confounded with non-tumor-related cfDNA alterations, as has been described by Leal et al. [32]. In Cohort 3, normal subjects were those in which the endoscopic evaluation of the stomach did not find any lesions that justify the performance of a tissue biopsy. Due to ethical restrictions, we do not have a histological assessment of these subjects. The predominance of women in the control and AG groups (81.7% and 69.1%, respectively) and, conversely, the men in the aGC group are other limitations of our results. In this study, we did not include OLGA staging to evaluate the premalignant conditions. We did not have sufficient data with the biopsy Sydney protocol to perform this analysis.

Our data show that methylated *RPRM* cfDNA should be considered a direct biomarker for the non-invasive detection of GC. Our results also highlight the utility of cfDNA as a predictive biomarker of patient treatment outcomes. The non-invasive nature of this approach will impact the compliance and participation rates of GC screening programs [42]. A clinical trial in an asymptomatic population will be necessary to evaluate the clinical performance of this biomarker and its cost-effectiveness as a screening strategy in GC.

## 4. Methods

### 4.1. Clinical Samples

*RPRM* promoter methylation was investigated in de-identified matched tumor and plasma samples and three prospectively collected series of GC, premalignant conditions, and healthy donors (HDs). De-identified samples were used for bisulfite sequencing, targeted sequence, and MethyLight development. The first prospectively collected series (Cohort 1, see Appendix A) comprised 81 participants with GC and 137 HD controls. The GC cases, confirmed by EGD and targeted biopsies, were accrued from the Gastric Cancer Prevention Center at Centro Referencia Salud (CRS), Servicio Salud Metropolitano Sur Oriente (SSMSO), Santiago [43] and from the Instituto Chileno–Japonés de Enfermedades Digestivas del Hospital Clinico San Borja Arriarán (ICHJED-HCSBA), Salud Metropolitano Sur Oriente (SSMC), Santiago, Chile. The 137 HDs were blood donors who received a health check at blood bank units at SSMSO and SSMSC. Blood samples from the cases and controls were collected from September 2004 to November 2008. Controls were accrued within two weeks of GC case collection. The second prospectively collected series (Cohort 2, see Appendix A) was composed of 27 samples from GC patients undergoing total gastrectomy, palliative chemotherapy, or perioperative chemotherapy from the Hospital Clinico de la Universidad Católica at Pontificia Universidad Católica de Chile (HCUC/PUC), Santiago, Chile. Patients from this cohort were grouped into responders (including partial responders, *n* = 17) and non-responders (including disease progressors, *n* = 10) according to the RECIST score [44]. The methylated *RPRM* cfDNA levels were analyzed at the beginning of treatment (T0) and again three months after (T1). Blood samples were collected from October 2009 to August 2010. A third population-based series of 1105 patients (Cohort 3, see Appendix A) referred for EGD was established at CRS, SSMSO, and ICHJED-HCSBA, SSMC, Santiago, Chile, between April 2010 and August 2013. All patients underwent EGD, and paired gastric tissues and plasma samples were collected. The series included 180 cases with normal stomachs, 692 cases diagnosed with AG, 140 with GIM, 13 with LGD, 3 with HGD, 18 with stage I and II eGC, and 59 with stage III and IV aGC. Cases diagnosed with GIM and LGD were grouped due to the small number in the latter group and their similar biological behavior (LGD = 13). HGD (*n* = 3) and eGC (*n* = 18) were grouped due to the small number of patients diagnosed in both groups and the similarities in their biological behavior [45]. All diagnoses were based on a histopathological analysis, except for normal stomachs, which were based on an EGD examination. Cases with LGH, HGD, and eGC were reviewed and confirmed by a second pathologist. We performed a nested case-control study (*n* = 111) in a subset of samples to evaluate the combined use of the methylated *RPRM* cfDNA and the pepsinogens (PG)-I/II ratio for the non-invasive detection of eGC. This design included 19 HGD/eGC cases and 3 to 6 age- and sex-matched premalignant conditions (92 AG/GIM/LGD). The EGD and histopathology reports (Hematoxylin and Eosin (H&E) and Giemsa stains) from both sites were abstracted onto study forms and entered into a Microsoft (MS) Access database [46]. Laboratory staff performing the testing were blinded to any characteristics of the samples. Ethical approval was obtained from the Internal Review Board of the Servicio Salud Metropolitano Sur Oriente and the Hospital Clinico Pontificia Universidad Católica de Chile.

### 4.2. Molecular Biology Methods

DNA extraction was performed from 1 mL of plasma obtained from subjects using the QIAamp DNA Mini Kit (QIAGEN, Germantown, MD, USA) following the instructions from the manufacturer. The DNA samples were resuspended in nuclease-free water and stored at −80 °C until use. According to the manufacturer’s instructions, bisulfite conversion was performed from 20 μL of extracted genomic DNA with the EZ DNA Methylation-Gold kit (Zymo Research, Irvine, CA, USA). Bisulfite sequencing of the *RPRM* promoter region was performed from bisulfite-treated DNA as previously described [17]. After cloning, the *RPRM* promoter region was sequenced using universal M13 primers by Macrogen (http://dna.macrogen.com (accessed on 29 November 2014)). Based on bisulfite sequencing results, we developed a fluorescence-based real-time PCR assay based on our previously reported methylation-specific PCR (MSP) [14]. Primers and probe design are available on request. Using a hybridization probe and a standard curve based on dilutions of known absolute quantities of the synthetic template allowed for the calculation of the absolute number of bisulfite-converted DNA molecules. In addition, this system has an increased sensitivity due to the exclusive discrimination of the methylated DNA after sodium bisulfite conversion [47]. To confirm the gastric origin of the plasma isolated samples, we performed a targeted sequence of PCR amplicon products using Ion Torrent technology (ThermoFisher, Waltham, MA, USA). Reads generated by fastq files were clusterized by the mean shift clustering algorithm (MeShClust) (v3.0, Kingsville, TX, USA) [48] and reads ≥ 1% were collected into a JavaScript Object Notation (JSON) database (Ecma International, Geneva, Switzerland). The Identity package (threshold = 0.8) was used to obtain the best score, and the makeTree.py script was used to plot the distance tree. Sequences were aligned with clustalW for each patient separately, and visualization was generated by Jalview software (v2, University of Dundee, Dundee, UK) [49].

### 4.3. Enzyme-Linked Immunosorbent Assay (ELISA) Method for the Detection of PG and H. pylori

Following manufacturer instructions, EDTA plasma samples were assayed for the PG-I/II levels using a commercial ELISA assay (Biohit, Helsinki, Finland). Plasma samples were tested singly with replicates across plates to evaluate assay reproducibility. Based on the manufacturer’s recommendations, the cut-off of the PG-I/II ratio was <3 [50]. A PG-I/II ratio lower than 3 indicates advanced corpus atrophy. Circulating whole-cell immunoglobulin G antibodies anti-*H. pylori* were measured using a commercial ELISA assay (Biohit, Helsinki, Finland) following the manufacturer’s instructions. The cutoff was defined as values ≥ 30 EIU.

### 4.4. Biostatistical Methods

For Cohort 1, the mean value of the methylated *RPRM* cfDNA level in the participants with GC and HD controls was compared using the Mann–Whitney test. Age differences between the participants with GC and HD controls were adjusted by quantile regression, and a pairwise comparison between both groups was performed. For a receiver operating characteristics (ROC) curve analysis, a training–validation approach in a blinded fashion was performed to identify the test’s best sensitivity and specificity. The area under the curve (AUC) was determined in the training cohort to identify the optimal cut-off value of the methylated *RPRM* cfDNA, which was evaluated in the validation cohort. For the treatment response (Cohort 2), levels of the methylated *RPRM* cfDNA were compared at T0 and T1 by the Wilcoxon matched pairs signed rank test. To evaluate the role of the methylated *RPRM* cfDNA in the stepwise progression of the precancerous cascade of the stomach (Cohort 3), we used sensitivity and specificity as well as negative and positive predictive and overall accuracy of the methylated *RPRM* cfDNA alone or in combination with PG. STATA 17 (College Station, TX, USA), R 4.4.1 (R Core Team 2024, Vienna, AU), and GraphPad Prisma 5 (Boston, MA, USA) software were employed.

## Figures and Tables

**Figure 1 ijms-26-03333-f001:**
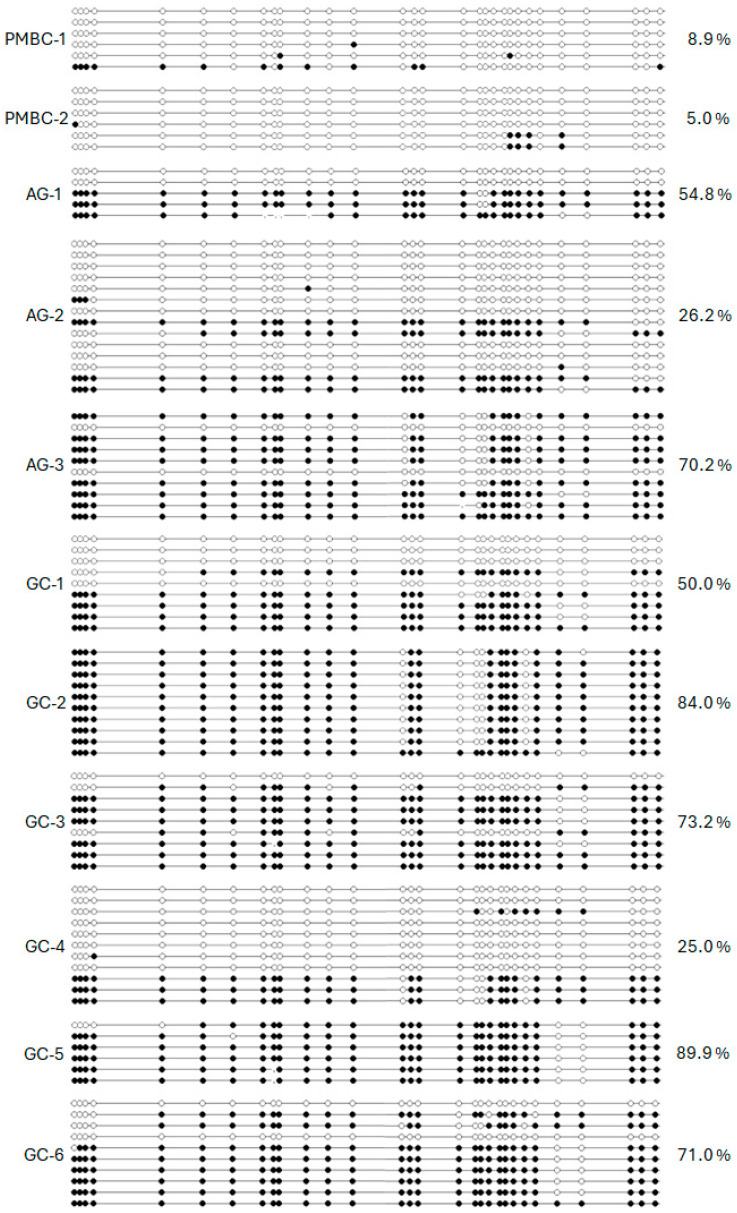
Bisulfite sequencing results of 30 CpG sites from −207 to +309 relative to the TSS of the promoter region of *RPRM* in peripheral blood mononuclear cells (PBMC-1 and -2) and stomach tissues including chronic atrophic gastritis (AG-1, -2, and -3) and gastric cancer (GC-1 to GC-6). White dots stand for unmethylated CpG sites and black dots stand for methylated CpG sites.

**Figure 2 ijms-26-03333-f002:**
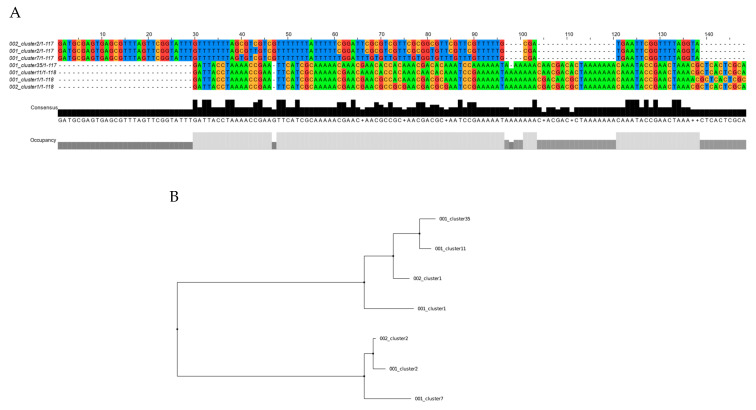
Targeted sequencing of the PCR amplicon products from paired tumor and plasma samples from a gastric cancer case. (**A**) Alignment of tissue samples (clusters 002) and plasma samples (clusters 001) by clustalW. (**B**) Distance trees of the same tissue and plasma clusters by the makeTree.py script. Two of five clusters were found in tumor and plasma samples (cluster1 and cluster2).

**Figure 3 ijms-26-03333-f003:**
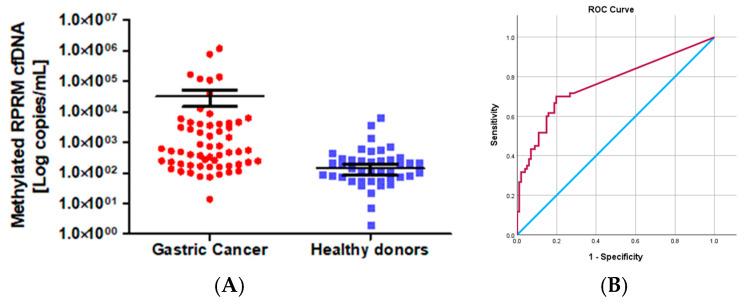
*RPRM* promoter methylation as a biomarker for non-invasive assessment of gastric cancer (Cohort 1). (**A**) Levels of methylated *RPRM* cfDNA in 81 cases of gastric cancer (mean value 32,240.00 copies/mL, SD =157,700.00 copies/mL) and 137 healthy donor controls (mean value139.00 copies/mL, SD = 657.00 copies/mL). Levels are significatively different by the Mann–Whitney test. (**B**) Receiver operating characteristic (ROC) curve for methylated *RPRM* cfDNA in plasma to determine the best cut-off point for sensitivity and specificity for non-invasive detection of GC. The light blue line represents the ROC curve. The red line represents the non-discrimination line between good and bad classification. The ROC curve for methylated *RPRM* cfDNA in plasma shows an area under the curve (AUC) of 0.764 (95% CI = 0.683–0.845) to determine the optimal cut-off value of 87.37 copies/mL (sensitivity 70.0%, and specificity 80.2%).

**Figure 4 ijms-26-03333-f004:**
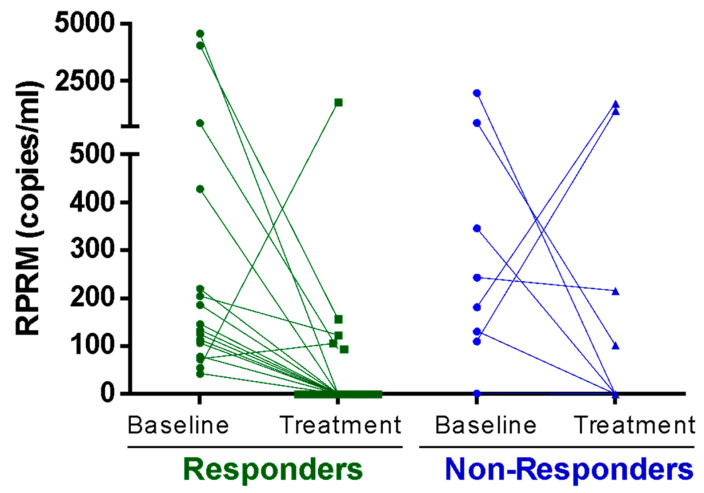
Treatment Response. Methylated Reprimo (*RPRM*) cell-free DNA (cfDNA) as a tumor marker for monitoring response to treatment (Cohort 2). Methylated *RPRM* cfDNA at initial diagnosis (baseline) and 3 months after starting treatment (treatment). Data was plotted separately for treatment-responder (e.g., disease free and partial response) and non-responder (stable and progressive disease) groups according to the Response Evaluation Criteria in Solid Tumours (RECIST). A statistically significant decrease in methylated *RPRM* cfDNA in plasma was observed in the responder group (*p* = 0.004, Wilcoxon matched pairs signed rank test), while no difference was observed in the non-responder group (*p* = 0.547).

**Table 1 ijms-26-03333-t001:** Overall evaluation of methylated *RPRM* cell-free DNA as a non-invasive diagnostic tool in gastric cancer (Cohort 3).

Statistic	Value	95% CI
Sensitivity	65.00%	53.52% to 75.33%
Specificity	75.90%	73.16% to 78.49%
Positive Likelihood Ratio	2.70	2.22 to 3.28
Negative Likelihood Ratio	0.46	0.34 to 0.62
Disease prevalence	7.24%	5.78% to 8.93%
Positive Predictive Value	17.39%	14.78% to 20.36%
Negative Predictive Value	96.53%	95.36% to 97.40%
Accuracy	75.11%	72.45% to 77.64%

**Table 2 ijms-26-03333-t002:** Methylated *RPRM* cell-free DNA detection by lesion grade (crude and adjusted by sex and age) (Cohort 3).

	Non-Adjusted				Adjusted by Sex and Age		
Lesion Grade	Positive *n* (%)	Odds Ratio (Crude Rate)	95% C.I. OR	*p*-Value	Odds Ratio (Adjusted)	95% C.I. OR	*p*-Value
Normal (No histological analysis was performed)	34 (18.89)	reference	reference	reference	reference	reference	reference
Chronic atrophic gastritis	167 (24.13)	1.37	0.91 to 2.06	0.164	1.37	0.90 to 2.07	0.140
Intestinal metaplasia and low grade dysplasia	46 (30.07)	1.85	1.11 to 3.07	0.020	1.60	0.93 to 2.75	0.091
Early gastric cancer (stage I-II) and high-grade dysplasia	10 (47.62)	3.90	1.53 to 9.94	0.004	3.23	1.15 to 9.11	0.026
Advanced gastric cancer (stage III-IV)	42 (71.19)	10.61	5.40 to 20.85	<0.0001	9.79	4.43 to 21.61	<0.0001
Chronic atrophic gastritis and gastric intestinal metaplasia/low-grade dysplasia	213 (25.21)	1.45	0.97 to 2.17	0.084	1.40	0.93 to 2.11	0.104
Early gastric cancer (stage I-II)/high-grade dysplasia and advanced gastric cancer	52 (65.00)	7.98	4.41 to 14.41	<0.0001	6.93	3.48 to 13.81	<0.0001

**Table 3 ijms-26-03333-t003:** Sensitivity, specificity and odds ratio of the combined use of methylated *RPRM* cfDNA and PG-I/II ratio for non-invasive assessment of early gastric cancer (eGC) (Cohort 3).

Test	Cases	Controls	Sensitivity		Specificity		Odds Ratio		
	*n* = 19	*n* = 92	%	95% CI	%	95% CI	Odds Ratio	95% CI	*p*-Value
Methylated *RPRM* cfDNA			47.37%	24.45% to 71.14%	77.17%	67.25% to 85.28%	3.04	1.09 to 8.47	0.033
positive	9	21							
negative	10	71							
PG-I/II ratio			52.63%	28.86% to 75.55%	80.22%	70.55% to 87.84%	4.51	1.60 to 12.72	0.005
Positive	10	18							
Negative	9	74							
Combined Methylated *RPRM* cfDNA or PG-I/II ratio			78.95%	54.43% to 93.95%	63.04%	52.34% to 72.88%	6.40	1.96 to 20.85	0.002
Positive	15	34							
Negative	4	58							

## Data Availability

Data is unavailable due to ethical restrictions.

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
