# Peer review of "Methylated Reprimo Cell-Free DNA as a Non-Invasive Biomarker for Gastric Cancer"

_ijms, 2025, doi:10.3390/ijms26073333_

Round 1
Reviewer 1 Report
Comments and Suggestions for Authors
The current study stands a chance to represent a significant advancement in the field of gastric cancer diagnostics, offering altogether a promising, potential alternative to traditional screening methods. By means of investigating the potential of methylated Reprimo cfDNA as a marker, the authors provide multiple evidence for its application in early detection and follow-up for treatment progression.
The study’s design aims at bias prevention from the first place, incorporating multiple patient cohorts with different characteristics and treatment plans and goals, and then furthers employs quantitative methylation-specific PCR to analyze plasma samples. The study statistical power is further strengthened by a well-structured presentation of data in an organized pattern. The combined methylated RPRM cfDNAand PG-I/II ratio 50 reached a sensitivity of 78.9% and specificity of 63.04%, demonstrating how it should be strongly considered as a solid marker for the non-invasivedetection of gastric cancer and for prediction of response to treatment.
Some limitations are worth to be mentioned. The absence of data concerning patients’ various comorbidities is among them, which could affect cfDNA levels and potentially produce a confounding effect, even though that is yet to be proven. Moreover, the analysis could be further reinforced by estimating or disclosing – if calculated - the rate of false positives and false negatives. Also, a comparative discussiontogether with other well-established gastric tumoral markerswould place into a better perspective the advantages and potential limitations of this novel marker in a broader therapeutic algorithm.
Even though minor adjustments would further enhance its clarity and impact, the research is overall well-executed, with results presented in-extenso and with sufficient transparency, standing on solid grounds for a meaningful contribution to the field. While I still recommend the paper for publication, further research for establishing feasibility in terms of clinical applications is warranted.
Author Response
We appreciate your positive comments. Regarding the issues you raised, we have responded point by point:
Comment 1: The absence of data concerning patients’ various comorbidities is among them, which could affect cfDNA levels and potentially produce a confounding effect, even though that is yet to be proven.
Answer: Thank you for pointing this out. We agree with your comment about the potential influence of comorbidities on cfDNA levels. However, Good Clinical Practice guidelines of our IRB participant institutions imposed on us the de-identification of the participants which does not permit access to the comorbidities data. This is something that we will request for future projects since it is becoming a critical factor in the reliability of cfDNA research.
Comment 2: Moreover, the analysis could be further reinforced by estimating or disclosing – if calculated - the rate of false positives and false negatives.
Answer: We appreciate and agree with this insight. In the revised version we have included this information in lines 322-323 “In this series we found 34 (69.4%) false positives and 4 (6.8%) false negatives”.
Comment 3: Also, a comparative discussion together with other well-established gastric tumoral markers would place into a better perspective the advantages and potential limitations of this novel marker in a broader therapeutic algorithm.
Answer: Thank you for this useful observation. In the revised version of the manuscript, we have included a paragraph (lines 370-387) with this discussion.
Reviewer 2 Report
Comments and Suggestions for Authors
The authors investigated methylation of the tumor suppressor gene RPRM in cfDNA as a biomarker for gastric cancer. They conducted a study using a previously reported fluorescence-based real-time PCR method. A derivation cohort (Cohort 1) consisted of 81 gastric cancer patients and 137 healthy controls to derive the cut-off value for methylated RPRM cfDNA. A group of 27 gastric cancer patients (Cohort 2) was used to examine the changes in methylated RPRM cfDNA and the therapeutic effect before and after treatment. A group of 1,105 patients (Cohort 3) was used to verify the sensitivity and specificity of the method. The cutoff value was calculated to be 87.37 copies/mL. The methylated RPRM cfDNA levels were significantly decreased in patients who responded to treatment, and they reported that the sensitivity and specificity of this examination for gastric cancer were 65% and 75.9%. The methods of this study were appropriate and the results were clearly presented. The study provides an important contribution to non-invasive detection of gastric cancer. However, the following some issues need authors' attention.
1. They conducted a study using three different cohorts, but it is difficult to tell which cohort each figure and table represents. They should state which cohort the results are from in the figure legends and table titles.
2. The results will be easier to understand if you add subheadings for each cohort.
3. They should provide more details about the RPRM genes in the introduction, including their functions and the effects of methylation.
Author Response
We appreciated your insights about our manuscript. Regarding your suggestions, we answer them point by point.
Comment 1: They conducted a study using three different cohorts, but it is difficult to tell which cohort each figure and table represents. They should state which cohort the results are from in the figure legends and table titles.
Answer: Thank you for pointing this out. In the revised version of the manuscript, we stated the cohort number in all figure legends and all table titles.
Comment 2: The results will be easier to understand if you add subheadings for each cohort.
Answer: Thank you for your comment. We added the name of each cohort to the subheadings of each results section.
Comment 3: They should provide more details about the RPRM genes in the introduction, including their functions and the effects of methylation.
Answer: Thank you for raising this point. In the revised version of the manuscript, we added a paragraph about the biology and pathology of RPRM genes to the introduction (see lines 76-80).
Reviewer 3 Report
Comments and Suggestions for Authors
The current manuscript investigates the use of methylated Reprimo (RPRM) cell-free DNA as a non-invasive biomarker for detecting gastric cancer (GC) and monitoring treatment response. The study involved three cohorts, including GC patients, individuals undergoing treatment, and population-based participants with varying gastric conditions. Using a real-time PCR assay, the research found that methylated RPRM cfDNA levels were significantly higher in GC patients compared to healthy controls and decreased in responders undergoing treatment. The biomarker demonstrated moderate sensitivity and specificity in distinguishing GC from premalignant conditions, suggesting its potential utility for early detection and therapeutic monitoring of GC.
The manuscript is well-structured and presents a significant advancement in non-invasive diagnostics for gastric cancer (GC) using methylated Reprimo (RPRM) cell-free DNA as a biomarker. However, I have some comments:
- All the (not sown data) should be provided as supplementary data.
- Most of the results are not represented in tables nor figures. It would be better to represent all the results in tables and figures rather than providing most of the data only in the text.
- The manuscript should be revised and typos should be corrected.
Comments on the Quality of English Language
Typos in the manuscript should be corrected.
Author Response
Thank you for your positive comments. The concerns that you raised have been addressed point by point:
Comment 1: All the (not shown data) should be provided as supplementary data.
Answer: Thank you for pointing this out. In the revised version of the manuscript, we added results described as “data not shown” as supplementary data. In the case of the section “RPRM promoter methylation in paired tissue and plasma gastric cancer samples”, we deleted “data not shown” since its main findings are shown in Fig 2, Fig S2 and Fig S3.
Comment 2: Most of the results are not represented in tables nor figures. It would be better to represent all the results in tables and figures rather than providing most of the data only in the text.
Answer: We appreciate your suggestion. In the revised version of the manuscript, we added new tables as supplementary information (see Supplementary Tables 1-5).
Comment 3: The manuscript should be revised, and typos should be corrected.
Answer: The typos have been corrected in the revised version of the manuscript.